# Reduction in Mortality of Calves with Bovine Respiratory Disease in Detection with Influenza C and D Virus

**DOI:** 10.3390/ani12233252

**Published:** 2022-11-23

**Authors:** Duanghathai Saipinta, Tanittian Panyamongkol, Phongsakorn Chuammitri, Witaya Suriyasathaporn

**Affiliations:** 1Department of Food Animal Clinic, Faculty of Veterinary Medicine, Chiang Mai University, Chiang Mai 50100, Thailand; 2Chiangmai Artificial Insemination and Biotechnology Research Center, Muang, Chiang Mai 50300, Thailand; 3Department of Veterinary Biosciences and Public Health, Faculty of Veterinary Medicine, Chiang Mai University, Chiang Mai 50100, Thailand; 4Research Center of Producing and Development of Products and Innovations for Animal Health and Production, Chiang Mai University, Chiang Mai 50100, Thailand; 5Center of Elephant and Wildlife Health, Chiang Mai University, Chiang Mai 50100, Thailand; 6Asian Satellite Campuses Institute-Cambodian Campus, Nagoya University, Nagoya 464-8601, Japan

**Keywords:** bovine respiratory disease, bovine viral diarrhoea virus, influenza C virus, influenza D virus, culling, reproductive performance

## Abstract

**Simple Summary:**

Respiratory diseases in cattle are multifactorial diseases caused by many pathogenic viruses and bacteria that cause economic losses both directly and indirectly. The objectives of this study were to assess the dead or culling and reproductive performances in heifer calves with different isolated viruses of bovine respiratory disease. We collected nasal swabs from calves that had respiratory signs in Chiang Mai Province, Thailand, and data with respect to age, farm type, and respiratory signs were recorded on the date of sample collection; thereafter, the data on the dead or culling, mortality, and reproductive performance of the BRD calves were collected. This study shows that the virus that causes the respiratory disease mixes with bovine viral diarrhoea virus (BVDV), influenza C (ICV), and influenza D virus (IDV). The calves with mucous secretion and at a younger age had higher risks of culling or death. On the other hand, the calves with IDV had a lower risk of culling. For the reproductive efficiency of BRD calves, we did not find any relationship between individual virus exposure and reproductive performance.

**Abstract:**

Both influenza C (ICV) and influenza D (IDV) viruses were recently included as bovine respiratory disease (BRD) causes, but their role in BRD has not been evaluated. Therefore, the mortality and reproductive performances of BRD calves with different isolated viruses were determined in this study. Data on 152 BRD calves with bovine viral diarrhoea virus (BVDV), bovine respiratory syncytial virus (BRSV), bovine coronavirus (BCoV), bovine parainfluenza virus 3 (BPIV-3), ICV, or IDV from nasal swab samples using real-time rt-PCR were used. The general data and respiratory signs were recorded immediately, and thereafter, the data on dead or culling calves due to BRD and reproductive performance were collected. The percentages of the BRD calves were 71.7%, 52.6%, 40.8%, 10.5%, 68.4%, and 65.8% for BVDV, BRSV, BCoV, BPIV-3, ICV, and IDV, respectively. Mucous secretion (OR = 4.27) and age ≤ 6 months (OR =14.97) had higher risks of mortality than those with serous secretion and older age. The calves with IDV had lower risks of culling than those without IDV (OR = 0.19). This study shows that most viral infections in BRD calves are a combination of viruses with BVDV, ICV, and IDV. In addition, IDV might have a role in reducing the severity of BRD calves.

## 1. Introduction

The longer raising of heifers causes unforeseen cost losses related to investment, the value of the animal, the opportunity cost of other management operations, and costs incurred by heifers that have ineffective reproductive systems, which commonly range from USD 1700 to USD 2400 [1]. Bovine respiratory disease (BRD) is a multifactorial disorder caused by many viral and bacterial infectious pathogens affecting the respiratory system [2]. The productivity impairment caused by BRD in dairy heifers includes decreased growth rates, increased culling rate, increased age at first service and first calving, and subsequently poor performance and reduced milk production in the first lactation [3,4,5]. 

The most common viral pathogens of BRD are bovine viral diarrhoea virus (BVDV), bovine coronavirus (BCoV), bovine respiratory syncytial virus (BRSV), bovine herpesvirus-1 (BHV-1), bovine parainfluenza virus 3 (BPIV-3), influenza C virus (ICV), and influenza D virus (IDV) [6,7,8]. BVDV causes chronic and persistent infection by establishing itself in immuno-privileged sites, such as the respiratory tract, ovaries, and testes, which affects reproductive performance in mature cattle [9,10]. BPIV-3 causing asymptomatic to severe pneumonia can predispose the animal to other viruses and bacterial infections, and the subsequent infections are important components of enzootic pneumonia and BRD in calves and heifers [11]. Both influenza C and influenza D viruses have been recently included as BRD causes in North America [7,12,13], Europe [14,15], and East Asia [16,17]. Although infection with ICV and IDV causes only mild respiratory disease in infected calves, the impact on the economy and productivity of dairy cows is in doubt. 

The effective control of BRD requires a combination of definitive diagnosis, efficacious BRD vaccines, appropriate treatment, and proper management practice [18]. In many tropical countries, including Thailand, control and eradication programs related to BRD have not been officially introduced to dairy farmers. In a report by the Bureau of Biotechnology on Livestock production from 2019 to 2021, the index of reproductive efficiency in replacement heifers in Thailand remained low, and the age of the first service was from 23.9 to 24.2 months, the age of first calving was from 33.6 to 33.8 months, and the first service conception rate was 26.8% [19], which might be due to tropical crossbred heifers reaching puberty later than purebred heifers in temperate regions [20]. However, information on the BRD-causing virus affecting the performances of heifers in this area is limited, and BRD might be an additional cause of low reproductive performance.

Therefore, the mortality of BRD calves in outbreak farms, i.e., the calves aged less than one year that exhibited respiratory signs related to the different viruses isolated, was determined in this study. In addition, the relationship between viruses and reproductive performance using days to first insemination and days to pregnancy was evaluated. 

## 2. Materials and Methods

### 2.1. Farm Selection, Samples, and Data Collection 

All animal care and sample collections were approved by the Institutional Animal Care and Use Committee at the Faculty of Veterinary Medicine, Chiang Mai University (approval no. S3/2565). 

This study was conducted from December 2020 to June 2021 using smallholder dairy farms with outbreaks of respiratory syndromes in calves aged 1 month to 1 year old in Maeon, Doiloh, and Chaiprakan Districts, Chiang Mai Province, Thailand. The farms which voluntarily participated in the study had 10 to 40 milking cows with mostly crossbred Holstein Friesian, and all the farms participated in the veterinary herd health management program of the Department of Livestock Development, Ministry of Agriculture and Cooperatives for more than five years. The BRD calves were included in this study if they exhibited respiratory signs, such as sneezing, coughing, fever (body temperature > 103°F), and serous or mucous nasal secretion. The outbreak of BRD on the farm was determined when there were more than two BRD calves at the same time. After an outbreak, the farmers informed a researcher (D.S.) and asked her to collect nasal secretion samples from the BRD calves using nasal cavity swabs. Subsequently, the swab samples were inserted into a transport tube containing viral transport media, kept on ice, and transported to the laboratory of the Faculty of Veterinary Medicine, Chiangmai University, within 24 h; then, they were kept at −80 °C until use. During the six months after sample collection, the participating farmers were responsible for reporting the mortality of the BRD calves, including the calves that died or were culled for reasons associated with BRD. The data on age, farm type, and respiratory signs were recorded on the date of sample collection. In May 2022, the follow-up data on mortality and the reproductive performance of the BRD calves were collected from the cow history sheets of the farms.

### 2.2. RNA Extraction and cDNA Synthesis

Viral RNA was extracted from a separate aliquot of each sample using a NucleoSpin^®^ RNA virus kit and cDNA synthesis by using a ReverTra Ace^®^ qPCR RT master mix, following the manufacturer’s instructions. The cDNA amplification reaction system was set at 20 μL, including 5 × RT Master Mix 4 at μL, and RNA template at 2 µg, and nuclease-free water was added to make the final volume 20 μL. The cDNA synthesis cycle was programmed as follows: 37 °C for 15 min, 50 °C for 5 min, 98 °C for 5 min, and finally held at 4 °C. The cDNA product was stored at −20 °C for use in the real-time rt-PCR.

### 2.3. Real-Time RT-PCR for Virus Detection

The cDNA product of each sample was tested for bovine viral diarrhoea virus (BVDV), bovine respiratory syncytial virus (BRSV), bovine parainfluenza-3 virus (BPIV-3), bovine coronavirus (BCoV), influenza C virus (ICV), and influenza D virus (IDV) by performing real-time rt-PCR. Real-time rt-PCR assays were performed on a 7500 Fast real-time PCR System (Applied Biosystems, Foster City, CA, USA) using a SensiFAST™ SYBR^®^ Hi-ROX Kit according to the manufacturer’s instructions. Briefly, the reaction system used for PCR amplification was 20 μL, consisting of 2x SensiFAST SYBR^®^ Hi-ROX Mix 10 μL, 10 μM of each primer, 1 μL of template cDNA, and RNAse-free water at 20 μL. Primers were taken from a recent publication (Table 1). For BVDV, cycling conditions were 95 °C for 2 min, 40 cycles of 95 °C for 5 s, and 55 °C for 30 s. For BCoV and ICV, the cycling conditions were 95 °C for 2 min and 40 cycles of 95 °C for 5 s and 62 C for 30 s. For BRSV, the cycling conditions were 95 °C for 2 min, and 40 cycles of 95 °C for 5 s and 59 °C for 30 s. For BPIV-3, the cycling conditions were 95 °C for 2 min and 40 cycles of 95 °C for 5 s and 55 °C for 62 s. For IDV, the cycling conditions were 95 °C for 2 min and 40 cycles of 95 °C for 5 s, and 64 °C for 30 s. Testing positive for any respiratory viral diseases was determined when their CT values were 1–40, and the CT values of more than 40 were determined as negative [21].

### 2.4. Statistical Analysis

The data on cattle characteristics were described by numbers and percentages. The BRD calves without any detection of the defined viruses were excluded due to unknown causes of diseases. The ages of the calves with BRD were categorised into ≤ 6 months and 7–12 months. The associations between the viruses, including BVDV, BCoV, BRSV, BPIV-3, ICV, and IDV, and different percentages of calves with fever, types of secretion (serous or mucous), and diarrhoea in the viral diseases were determined using Fisher’s Exact Chi-square test.

Mortality was determined when the BRD calves died or were culled from their severe illness for reasons associated with BRD within six months after infection. The associations of ages, fever, mucous secretion, diarrhoea, and viruses with mortality were described using Fisher’s Exact Chi-square test. The variables with *p* < 0.20 were selected for further analyses by using a generalised estimating equation, a method for modelling clustered binary data, using the Genmod procedure (SAS Institute Inc., Cary, NC, USA). The cattle nested in the same farm were considered as the repeated factors. To define the factors associated with mortality, variables with *p* < 0.1 were entered and subsequently stayed in the final model.

The reproductive performances of the BRD calves were determined using Kaplan–Meier survival statistics. The ages to the first artificial insemination (AI) and the ages to pregnancy were survival times in which the BRD calves that were not AI or pregnant within two years were censored cases, and their survival times were 730. For the other censored cases, the age at the end of follow-up, such as the age at the culling date or the age at the last AI but not pregnant, was the survival time. The estimated median values of the reproductive performances between the cattle with and without various viral diseases were described and compared using the generalised Wilcoxon test. The significance level and the tendency were defined at *p* < 0.05 and *p* < 0.10, respectively.

## 3. Results

Of the total of 168 BRD calves, 16 (9.5%) were excluded due to undefined viral causes of disease. The percentages of the BRD calves were 71.7%, 52.6%, 40.8%, 10.5%, 68.4%, and 65.8% for BVDV, BRSV, BCoV, BPIV-3, ICV, and IDV, respectively. Only 22 BRD calves (14.5%) had only one infection, including BVDV (10/109) and BRSV (5/80), which had the highest single infection rates. The values of the highest combination infection prevalence were 12.5% (*n* = 19) for BVDV + BRSV + BCoV + ICV + IDV, 9.9% (*n* = 15) for BVDV + BRSV + ICV + IDV, 8.6% (*n* = 13) for both BVDV + BCoV + ICV + IDV and BVDV + ICV + IDV, and 7.2% (*n* = 11) for ICV + IDV. Differences in the percentages of respiratory signs of the viral diseases are shown in Table 2, including fever (7.9%), diarrhoea (55.3%), and mucous secretion (67.8%).

The BRD calves with BVDV or BCoV had diarrhoea more than those without both diseases. The BRD calves with BVDV tended to have a higher proportion of the sign of the presence of mucous nasal secretion than those without BVDV (*p* = 0.056). In addition, the BRD calves with BCoV had a closed tendency of a higher percentage of mucous secretion than those without BCoV (*p* = 0.11). The percentages of pairwise diseases occurring together are shown in Table 3. Influenza D virus occurrence had chances of coinfection with either ICV (*p* < 0.01) or BCoV (*p* = 0.11). The BRD calves with BVDV were at a higher risk of being coinfected with BCoV (*p* = 0.10).

After infection, 18 BRD calves (11.8%) died or were culled within six months after incurring the disease, and the days ranged from 18 to 177 days with an average of 96.8 days. All BRD calves died or were culled from severe weakness problems, including severe respiratory problems (*n* = 11), bloating (*n* = 2), and comorbidity (*n* = 5). Comparisons of the mortality rates of the BRD calves with different signs and their detected viruses are shown in Figure 1. The age at BRD (*p* < 0.05), BVDV infection (*p* = 0.10), ICV (*p* < 0.05), and IDV (*p* < 0.05) were associated with culling rates. The calves aged ≤ 6 months had higher culling rates than older calves (Figure 1A), and the calves with BVDV tended to be culled more than those without BVDV (Figure 1B). In contrast, the calves with ICV and IDV had lower mortality rates (Figure 1B). 

The final model of factors associated with the mortality of BRD calves is shown in Table 4, including the type of nasal secretion, the detection of IDV, and the age of BRD. The calves with mucous secretion (OR = 4.27) and the BRD calves ≤ 6 months of age (OR = 14.97) had a higher risk of mortality than those with serous secretion and older age. The BRD calves with IDV had a lower risk of culling compared with those without IDV (OR = 0.19).

Results for the Kaplan–Meier analysis showed that, in terms of the median times of age, 50% of the BRD calves having the first AI and pregnancy were 572 and 710, respectively, as shown in Figure 2. The BRD calves with BVDV detection had a higher probability to have the first AI (*p* = 0.10) and pregnancy (*p* < 0.05) earlier than the calves without BVDV. The median ages of the BRD calves with and without BVDV were 572 and 521 days old for having AI and 683 and >730 days old for pregnancy, respectively. No other factor was found to be significant.

## 4. Discussion

Due to the significant economic impact of BVDV on cattle production, many countries, including Austria, Belgium, Denmark, England, Finland, Germany, Ireland, Norway, Poland, Scotland, Sweden, Switzerland, the Netherlands, the USA, and Wales, have implemented compulsory or voluntary control and/or eradication programs [27]. Monitoring with ELISA test kits showed that BVDV is a globally spreading pestivirus affecting cattle and other ruminants, for example, in Thailand [28] and China [29]. However, many countries in the world, including Thailand, do not have a national policy for the prevention and control of BVDV and other viral BRD, although BVDV was discovered in Thailand in 1997 [30]. The high number of clinical cases in our study does not fall outside our expectations. Without any BVDV management programs in Thailand, most cases of the respiratory syndromes observed in this study were likely associated with those calves that were persistently infected (PI) due to the expected BVDV infection during the first trimester of pregnancy [31]. In addition, the design of this study aimed to determine the subsequent association among the signs, viruses, mortality rate, and reproductive performance of the calves experiencing respiratory signs. Therefore, the studied population was only the BRD calves with the specified viral infection. Any application of the results of this study must be carefully considered and only extrapolated to the BRD population. 

Among the viruses causing BRD, all RNA viruses, including newly defined viruses such as ICV and IDV, were included in this study [13,32]; therefore, bovine herpesvirus-1, a double-stranded DNA virus, was excluded due to the limitations of the viral genome extraction method. The BRD has many causes and complexities, but controlling BVDV is important for suppressing secondary infections from other related BRD pathogens with respect to the immune system [33]. Therefore, mixed infections were expectedly found to be more frequent (85.5%), with the highest infections observed with BVDV (71.7%), ICV (68.4%), and IDV (65.8%) in this study. The patterns of mixed viral infections in our study were BVDV + BRSV + BCoV + ICV + IDV, BVDV + BRSV + ICV + IDV, BVDV + BCoV + ICV + IDV, BVDV + ICV + IDV, and ICV + IDV. In Mexico, mixed viral pathogens, including BVDV, BRSV, and BPIV-3, were identified in over 50% of the clinically diseased lungs of calves [34]. BRSV and BCoV also highly contributed to the cause of BRD calves in this study [35,36]. Clinical respiratory disease associated with BCoV in combination with BRSV has been reported in Danish dairy calves [37], while a study in the USA by Fulton et al. also found BCoV associated with BRSV and BoHV1 infections [38]. This is the first detected ICV and IDV in dairy cattle in tropical countries, with the prevalence of ICV and IDV at 68.4%, and 65.8%, respectively. Both ICV and IDV have been first detected in BRD cattle in North America [39,40]. As found in this study, the most common viruses found to be in coinfected with ICV and IDV were BVDV [7,41,42]. 

The symptoms in BRD cases vary depending on the virus cause, starting from no signs, mild respiratory signs, and severe systemic signs with fever, mucous discharges, and diarrhoea. In our study, 7.9%, 55.3%, and 67.8% of the BRD calves had fevers, mucous discharge, and diarrhoea, respectively. The BRD calves with BVDV and BCoV detected were associated with diarrhoea, and BVDV was also significantly associated with mucous nasal discharge (Table 2). BVDV is clinically presented in multiple organs, including those of the respiratory, gastrointestinal, and reproductive systems [43]. Diarrhoea in calves with BCoV infection is caused by necrotic enterocolitis with a loss of villi causing malabsorption diarrhoea [44]. 

From this study, 11.8% of the BRD calves died or were culled by severe respiratory problems (61%), other infectious diseases (28%), and digestive problems (11%). In 2020, BRD morbidity and mortality in the USA, where the control program has been intensively applied, were 18% and 2.1%, respectively [45], but this was not the case for the uncontrolled BRDs in this study. Our study’s results were within the range of the data from the USA from 1987 to 2001, in which mortality rates, including calves that died or were culled for reasons associated with the BRDs of feedlot calves, ranged between 0.1 and 8.9 depending on their breed [46]. In 2021, the mean cumulative BRD mortality incidence in beef calves ranged between 1 and 5 depending on the maximum temperatures and wind speeds [47]. The differences in this mortality might be caused by the fact that all the animals used in this study showed clinical respiratory signs. In Table 4, three factors, including the presence of mucous nasal secretion, the detection of IDV, and the age at sickness, were related to culling in these BRD calves. In BRD, serous nasal discharge is common in the early stages, and it turns mucopurulent as the condition progresses. As the disease progresses, dyspnoea becomes more pronounced and animals may adopt a typical stance with an extended neck, drooling with open-mouth breathing, and soft coughing. Effective mucus clearances are essential for lung health, and airway disease is a consistent consequence of poor clearance, causing pneumonia [48]. One of the highest mortality causes of BRD is severe pneumonia [49]. The virus detected from the nasal swabs in this study might not actually cause severe pneumonia of BRD related to the subsequent dead heifer calves. Primary viral infection impairing host defences and predisposing to secondary bacterial infection is preeminent pathogenesis [18]. Determining the actual causes of mortality, especially related to severe pneumonia, may need to define the pathogens from the lower respiratory tracts and lung samples collected via other collection methods, for example, post-mortem lung tissue specimens [6,13,34], pharyngeal swabs [7], and bronchoalveolar lavage fluid [8]. In this study, the BRD calves with symptoms at age ≤ 6 months had more mortality rates than older calves. In addition, many studies showed that younger calves indicated by no-weaned calves [47], and those aged between 1 day and 6 months [50] were more at risk for culling or death than weaned calves and older calves. 

In general, cattle that have higher viral loads of IDV and ICV also have greater numbers of coinfecting viruses than controls, suggesting that ICV and IDV may be significant contributors to BRDC [7]. This study is the first to determine the clinical role of IDV in BRD calves. Interestingly, IDV was found to decrease the risk of mortality in the BRD calves in this study. This is consistent with another study that used mice to show that IDV infections appeared to protect mice from the usual clinical features of secondary *Staphylococcus aureus* infections. IDV-infected mice improved their weight, survival rate, and recovery times compared with those infected only with *S. aureus* [51]. Infection with IDV decreases the susceptibility to secondary bacterial infection, as shown in the evidence that antiviral immune responses occurred after IDV infection to protect the host from potentially fatal outcomes. In addition, Robinson and others demonstrated that determining the immune response to IDV infection beneficially affected the prevention of the severity of the influenza A virus (IAV) [52]. IDV-IAV-infected mice showed faster and stronger IFN-β responses than IAV-infected mice alone. Moreover, their results show that active IDV infection induces IFN type 1-dependent protection against IAV-associated weight loss and mortality.

From Figure 2, it can be observed that the BRD calves had overall ages to the first AI and pregnancy of 572 (19 months) and 710 (23 months), respectively, and these would subsequently calve at the age of about 30 months. A recent review of UK dairy herd data showed that the average age at first calving was 29 months (median 28 months) [53], and this seemed significantly greater than the target of 22–24 months, which is often quoted in the press [54]. In contrast, a report using 68,555 cows in Thailand shows that the ages at first service and calving at 22.9 and 32.9 months, respectively [20], were higher than those observed in our report here. The participation in the veterinary herd health management program of all the farms in this study might be the reason for the better reproductive performance compared with other farms in Thailand, but the remaining poor reproductive performance might be related to the respiratory syndrome incurred during calfhood. In the current study, the occurrence of BRD with BVDV detection was probably associated with having the first AI and pregnancy at a younger age than in heifers without BVDV. In comparison to BRD-free calves, the calves with BVDV infections had more impaired reproductive performances [55,56], especially due to their harmful impact on the developing foetus at all stages, but more severe consequences occur during early gestation [57]. As all heifers had experiences with BRD during their calfhood, the reproductive performances of BRD calves with BVDV in this study were better than that of BRD calves with other virus infections.

## 5. Conclusions

In conclusion, this study is the first to investigate the role of ICV and IDV in BRD calves. We found that most viral infections in BRD calves were a combination of viruses with BVDV, ICV, and IDV, and factors such as the presence of mucous nasal secretion, the detection of IDV, and age at sickness were related to culling in BRD calves. In addition, IDV might play an important role in reducing the severity of other viral or secondary bacterial infections in BRD calves. Therefore, further research on IDV is needed and should include the pathophysiology and correlations with other viruses and/or bacterial infections in BRD calves.

## Figures and Tables

**Figure 1 animals-12-03252-f001:**
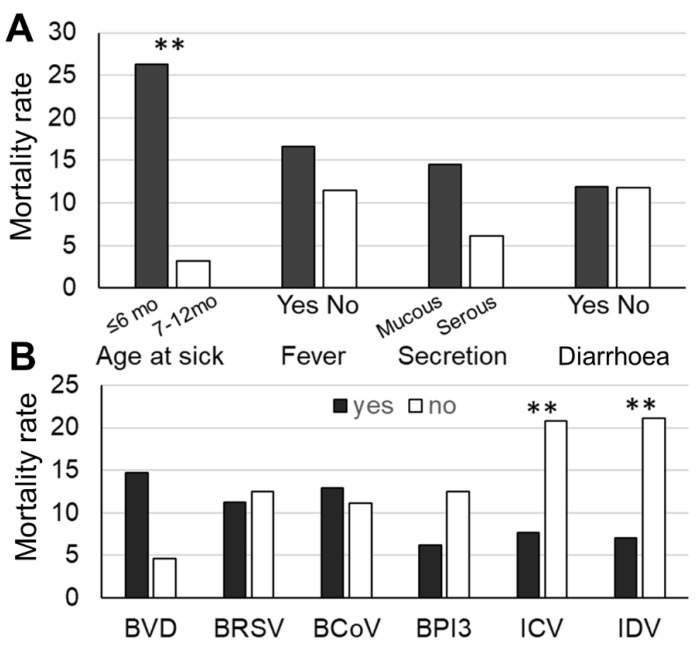
Comparisons of mortality rates of bovine respiratory disease (BRD) calves between with and without factors (**A**) and virus detection (**B**). ** indicate different culling rates at *p* < 0.05, respectively. BVDV: bovine viral diarrhoea virus; BRSV: bovine respiratory syncytial virus; BCoV: bovine coronavirus; BPIV-3: bovine parainfluenza virus 3; ICV: influenza C virus; IDV: influenza D virus.

**Figure 2 animals-12-03252-f002:**
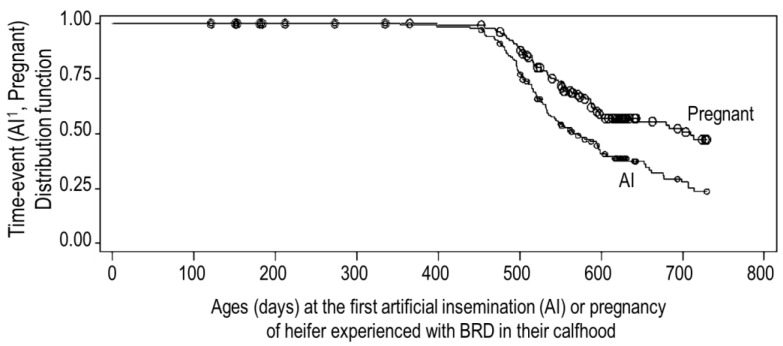
Kaplan–Meier distribution function of ages at the first artificial insemination (AI) and pregnancy of the bovine respiratory disease (BRD) calves.

**Table 1 animals-12-03252-t001:** Sequences of the primers used to detect viral pathogens in this study.

Type	Gene Fragment	Primer	Sequence (5′ to 3′)	Ref.
^1^ BVDV	Matrix	BVDV-F BVDV-R	GGGNAGTCGTCARTGGTTCG GTGCCATGTACAGCAGAGWTTTT	[22]
^2^ BRSV	N gene	BRSV-F BRSV-R	GCAATGCTGCAGGACTAGGTATAAT ACACTGTAATTGATGACCCCATTCT	[23]
^3^ BCoV	Matrix	BCoV-F BCoV-R	CTGGAAGTTGGTGGAGTT ATTATCGGCCTAACATACATC	[24]
^4^ BPIV-3	P gene	PI3-F PI3-R	AGAGCACTCRATTTACAGARAGG GTATCYGCATTGTTNAGGACATT	[25]
^5^ ICV	Matrix	ICV-F ICV-R	TCGGCAGATGGGAGAGATG GAATTGGTGAGTTGTCGGTTTC	[13]
^6^ IDV	PB1 gene	IDV-F IDV-R	TGGATGGAGAGTGCTGCTTC GCCAATGCTTCCTCCCTGTA	[26]

^1^ BVDV: bovine viral diarrhoea virus. ^2^ BRSV: bovine respiratory syncytial virus. ^3^ BCoV: bovine coronavirus. ^4^ BPIV-3: bovine parainfluenza virus 3. ^5^ ICV: influenza C virus. ^6^ IDV: influenza D virus.

**Table 2 animals-12-03252-t002:** Comparison of percentages of signs in the viral disease of bovine respiratory disease (BRD) calves, the <1-year calves with respiratory syndromes. * and ** indicate a difference between with (Yes) and without the disease (No) at *p* < 0.1 and *p* < 0.05, respectively.

	Fever		Diarrhoea		Mucous	
	Yes	No	Yes	No	Yes	No
^1^ Viral Disease	(*n* = 12)	(*n* = 140)	(*n* = 84)	(*n* = 68)	(*n* = 103)	(*n* = 49)
BVDV (*n* = 109)	8.3	6.0	64.2 **	32.6 **	72.5 *	55.8 *
BRSV (*n* = 80)	6.3	9.7	53.8	56.9	67.5	68.1
BCoV (*n* = 62)	4.8	10.0	80.7 **	37.8 **	75.8	62.2
BPIV-3 (*n* = 16)	6.3	8.1	50.0	55.9	68.8	67.7
ICV (*n* = 104)	5.8	12.5	58.7	47.9	69.2	65.6
IDV (*n* = 100)	7.0	9.6	59.0	48.1	71.0	61.5

^1^ BVDV: bovine viral diarrhoea virus; BRSV: bovine respiratory syncytial virus; BCoV: bovine coronavirus; BPIV-3: bovine parainfluenza virus 3; ICV: influenza C virus; IDV: influenza D virus.

**Table 3 animals-12-03252-t003:** Percentages of a virus (row) to infect together with (Yes) or without (No) another virus (column). * and ** indicate associations among both virus, row, and column, at *p* < 0.10 and 0.05, as a tendency and significant levels, respectively.

	BRSV		BCoV		BPIV-3		ICV		IDV	
	Yes	No	Yes	No	Yes	No	Yes	No	Yes	No
BVDV	73.8	69.4	79.0 *	66.7 *	75.0	71.3	70.2	75.0	70.0	75.0
BRSV	-		58.1	48.9	56.3	52.2	53.9	50.0	50.0	54.0
BCoV	-		-		37.5	41.2	43.3	35.4	46.0 *	30.8 *
BPIV-3	-		-		-		11.5	8.3	9.0	13.5
ICV	-		-		-		-		93.0 **	21.2 **

BVDV: bovine viral diarrhoea virus; BRSV: bovine respiratory syncytial virus; BCoV: bovine coronavirus; BPIV-3: bovine parainfluenza virus 3; ICV: influenza C virus; IDV: influenza D virus.

**Table 4 animals-12-03252-t004:** Factors associated with the mortality rate of bovine respiratory disease (BRD) calves.

				Odds Ratio	95% CI of Odds Ratio		Chi-Square	*p*
Parameter	Level	Estimate	SEM		Lower	Upper		
Secretion	Mucous	1.45	0.73	4.27	1.03	17.79	3.98	0.05
	Serous	-------------------------------Reference------------------------------						
Detection of IDV	Yes	−1.66	0.60	0.19	0.06	0.61	7.75	0.01
	No	-------------------------------Reference------------------------------						
Age at sick	≤ 6 months	2.71	0.70	14.96	3.79	59.05	14.91	>0.01
	7–12 months	-------------------------------Reference------------------------------						

IDV: influenza D virus.

## Data Availability

The data presented in this study are available upon request from the corresponding author.

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
