# Peer review of "Reduction in Mortality of Calves with Bovine Respiratory Disease in Detection with Influenza C and D Virus"

_animals, 2022, doi:10.3390/ani12233252_

Round 1
Reviewer 1 Report
The authors measured the mortality and reproductive performance of BRD calves with different virus isolates. Authors found that most viral infections in BRD calves were a combination of viruses with BVDV, ICV, and IDV. In addition, IDV might have a role in reducing the severity of BRD calves.
The manuscript contains interesting results, but I have several comments to improve the manuscript.
Section 1-Line no. 62-65: sentence is confusing. Please re-phrase it.
Line no. 73-77: "In a report by...", a reference is needed.
Section 2.1 in M&M, the authors mentioned that ''The study was conducted..using smallholder dairy farms..in ...", however, the number and size of smallholder cattle farms was not properly described. "fewer than 40 milking cows", this description is puzzling.
Section 2.4 is very long, it could be shorter and clearer than in the present status.
Authors should check carefully the nomenclature. (Section 2.4,Section 3 for instance) Sometimes authors wrote " p-value", sometimes "P".
Section 4 - Line no. 291-292: Wrong reference location(reference 50 for instance). Sentence is not well structured. Please re-word it.
Line no. 316, the authors mentioned they found that "BVDV, ICV, and IDV are the most mixed viruses". This sentence is confusing and needs to be re-word it.
Please update the references in a uniform format.
The English writing should be checked carefully.
Reviewer 2 Report
Dear Authors
This study reports on a field collection of nasal swab samples from calves showing signs of suspected BRD. I sincerely hope my comments are taken as a constructive criticism only. The manuscript thas some benefits but has a lot of work that has to be done
1. English needs significant improvement
2. Limitations of the study must be addressed
a. Nasal swabs are not always representative of the lung pathogens. There is plethora of literature saying this. Recognition of this limitation does not decrease the study value.
b. Negative controls not included (e.g., no 'healthy' calves were swabbed at the same time). This limitation is very important and may have put the study design in jeopardy. It must be addressed and study result should be clearly restricted to the studied population and not easily extrapolated to external population.
All my minor comments are included in the attached document (pdf).

Round 2
Reviewer 2 Report
Authors have addressed all my comments satisfctorily.
However, they are yet to address the comment about nasal swabs versus lung sample isolates (e.g., transtracheal wash or broncho-alveolar lavage).
Author Response
Authors have addressed all my comments satisfctorily.
However, they are yet to address the comment about nasal swabs versus lung sample isolates (e.g., transtracheal wash or broncho-alveolar lavage).
AU: Thank you for your valuable attention. In this study, the virus infection from nasal swab was treated as a risk factor for the mortality of BRD calves. To prevent the confusion of the readers, especially the lung pathogens, we agree to add more discussion on this.
Page 8 Line 299: Added the sentences: “The virus detected from nasal swabs found in this study might not actually cause severe pneumonia of BRD relating to the subsequent dead heifer calves. Primary viral infection impairing host defenses and predisposing to secondary bacterial infection is preeminent pathogenesis [18]. Determining the actual causes of mortality especially relating to severe pneumonia may need to define the pathogens from lower respiratory tracts and lung samples collected by other collection methods for example postmortem lung tissue specimens [6,13,34], pharyngeal swabs [7] and bronchoalveolar lavage fluid [8].”